# Identifying the lungs as a susceptible site for allele-specific regulatory changes associated with type 1 diabetes risk

Daniel Ho[1,7], Denis M. Nyaga[1,7], William Schierding[1,2], Richard Saffery[3], Jo K. Perry[1,2], John A. Taylor[2,4], Mark H. Vickers[1], Andreas W. Kempa-Liehr[5] & Justin M. O'Sullivan[1,2,4,6✉]

Type 1 diabetes (T1D) etiology is complex. We developed a machine learning approach that ranked the tissue-specific transcription regulatory effects for T1D SNPs and estimated their relative contributions to conversion to T1D by integrating case and control genotypes (Wellcome Trust Case Control Consortium and UK Biobank) with tissue-specific expression quantitative trait loci (eQTL) data. Here we show an eQTL (rs6679677) associated with changes to *AP4B1-AS1* transcript levels in lung tissue makes the largest gene regulatory contribution to the risk of T1D development. Luciferase reporter assays confirmed allele-specific enhancer activity for the rs6679677 tagged locus in lung epithelial cells (*i.e.* A549 cells; C > A reduces expression, $p = 0.005$). Our results identify tissue-specific eQTLs for SNPs associated with T1D. The strongest tissue-specific eQTL effects were in the lung and may help explain associations between respiratory infections and risk of islet autoantibody seroconversion in young children.

[1] Liggins Institute, The University of Auckland, Auckland, New Zealand. [2] The Maurice Wilkins Centre, The University of Auckland, Auckland, New Zealand. [3] Murdoch Children Research Institute, The University of Melbourne, Melbourne, Australia. [4] School of Biological Sciences, The University of Auckland, Auckland, New Zealand. [5] Department of Engineering Science, The University of Auckland, Auckland, New Zealand. [6] MRC Lifecourse Epidemiology Unit, University of Southampton, Southampton, UK. [7] These authors contributed equally: Daniel Ho, Denis M. Nyaga. ✉email: justin.osullivan@auckland.ac.nz

Type 1 diabetes (T1D) is characterised by immune-mediated destruction of insulin-producing pancreatic beta cells leading to loss of insulin production and hyperglycaemia. Population-level data has enabled genome-wide association studies (GWAS) that have identified ~60 genetic loci that are associated with the risk of developing T1D[1]. In addition to the GWAS studies, a number of highly phenotyped prospective birth cohort studies have investigated potential early determinants of T1D risk[2–4]. Notably, the transition from genetic risk to T1D onset is hypothesised to require an environmental trigger event, such as infection, in those individuals who go on to develop the disorder[5]. However, the mechanisms responsible for this transition remain poorly characterised, limiting strategies for optimising treatment and furthering therapeutic development.

One hindrance to characterising the genetic mechanisms responsible for T1D development is the finding that the majority of SNPs are within intergenic regions of the genome. Previously, we used information on the spatial organisation of the genome (captured by Hi-C) to identify the tissue-specific gene regulatory impacts (i.e. eQTLs) of SNPs associated with T1D[6]. Consistent with our understanding of T1D pathology, we reported that the differentially expressed genes were enriched for immune activation and response pathways[6]. However, this still did not provide any important information into the relative contributions of the tissue-specific gene regulatory effects we identified. Therefore, we reasoned that we could use genotypes for T1D cases and controls to machine learn the tissue-specific-expression scores for T1D-associated variants. This approach would enable the ranking of the tissue-specific regulatory changes that contribute to the conversion of genetic risk to T1D pathology.

In the present study, we assigned SNPs associated with T1D to the genes they modulate through Hi-C chromatin interactions captured from primary tissues (i.e. pancreas and spleen) and immortalised cells. We integrated a regularised logistic regression model on European ancestry genotypes of T1D case and control to identify transcriptional changes in the lung involving *AP4B1-AS1* and *CTLA4* (associated with rs6679677) as the largest individual contributors, through a gene regulatory mechanism, to the conversion of the genetic risk for the development of T1D. Finally, a plasmid-based luciferase reporter expression assay was performed to validate the allele-specific enhancer activity of the locus marked by rs6679677 in lung cells.

## Results

**T1D SNPs impact an extensive gene regulatory network**. The methodology used for the characterisation of the regulatory networks for T1D-associated SNPs is summarised in Fig. 1. Briefly, we used the CoDeS3D[7] algorithm to analyse 313 T1D-associated SNPs (Methods section and Supplementary Data 1) using Hi-C chromatin contact libraries (Supplementary Data 2) and GTEx (v7) RNAseq data (Methods section). The Hi-C libraries that were used in this study included immortalized cell lines and primary human tissues (Supplementary Data 2) and were chosen to ensure a range of known possible interactions were included in the analysis. We define a spatial eQTL as a SNP that tags a locus that: (1) physically interacts with a gene; and (2) explains a fraction of the genetic variance of the interacting gene transcription level. According to our definition, the eQTL variant can sit anywhere within the genome. This includes within the boundaries of the gene, as long as the gene is covered by ≥3 restriction fragments in the Hi-C library. This minimum connection distance is determined by Hi-C resolution, which cannot distinguish spatial connections between ligated contiguous restriction sites versus an undigested restriction site. Of the 313 SNPs, 57 SNPs had no identifiable eQTLs, resulting in 256 T1D-associated SNPs connecting

to 822 genes (1479 spatial eQTL-eGene associations; FDR $q < 0.05$; Supplementary Data 3). As expected from our previous study[6], the 822 genes were enriched for immune activation and response pathways (Supplementary Data 4).

The eQTL-eGene interactions were categorised as either: *cis*, the eQTL and eGene are separated by a linear distance of ≤1 Mb on the same chromosome; or *trans*, eQTLs and their eGenes were separated by >1 Mb on the same chromosome or located on different chromosomes. Notably, of the 256 T1D-associated SNPs with spatial-eQTLs, 190 affected the *trans*-regulation of 361 genes, while 201 affected the *cis*-regulation of 493 genes. Some genes ($n = 32$) were regulated by different eQTLs in both *cis* and *trans* (e.g. *TRIM26, RNF5, PSMB9* and *NOTCH4*; Supplementary Data 3). Notably, the 112 *trans*-regulated genes (e.g. *FOXP1*, *CAMTA1* and *ROBO2*) we identified are enriched for being less tolerant of inactivating (i.e. Loss-of-Function) mutations (Supplementary Fig. 1 and Supplementary Data 5).

**Machine learning identifies transcriptional changes in the lung as being involved in the conversion of risk to T1D**. Based on our eQTL analysis, we determined that T1D SNPs form an integrated gene regulatory network across tissues and immune cell types. We reasoned that we could use a machine learning approach to integrate the tissue-specific spatial eQTL-eGene associations with thousands of individual genotypes from large T1D cohorts to convert population-level risk (i.e. GWAS SNPs) to individualised risk (i.e. the burden for an individual's genotype).

We trained and validated the predictive accuracy of a regularised logistic regression predictor (Methods section and Supplementary Fig. 2), which predicts the T1D disease status. This model was used to estimate the additive tissue-specific contribution of the spatial eQTLs within genotypes from individuals who developed T1D (WTCCC; 1960 T1D cases and 2933 controls). Individual genotypes were weighted using the tissue-specific spatial eQTL effect sizes from the CoDeS3D analysis (Supplementary Data 3). Of the 313 T1D-associated SNPs, 253 were present within each of the WTCCC genotypes (Supplementary Data 6). Of the 253 SNPs, 224 had identifiable eQTLs, connecting to 758 eGenes (6307 tissue-specific eQTL effects).

Essential feature selection was performed using the Mann–Whitney $U$ test[8] (selected 2048 data features from 6307 eQTL features and 29 SNP features with unknown eQTL effects) and lasso regularisation of the logistic regression. The regularisation parameter was optimised by sampling 80% of the WTCCC derived eQTL dataset, identifying optimal hyperparameters from this sample, and evaluating the performance of the algorithm on the remaining 20% using the optimal hyperparameters. In order to identify optimal hyperparameters, the selected 80% of data were divided into 10 subsets, with each subset having approximately the same number of samples. For every hyperparameter value, the prediction model was trained repeatedly on 9 subsets and evaluated on the remaining subset until every subset had been used for evaluation once. The optimal regularisation parameter was selected based on the AUC (model 1). Applying this model to the validation data (20% of the WTCCC derived eQTL dataset) resulted in an AUC of 0.76, which is acceptable for our purpose of developing an interpretable machine learning model, which identifies the top-performing predictors from additive tissue-specific contributions of the spatial eQTLs within genotypes from individuals who developed T1D. In order to quantify the uncertainty of the model performance subjective to different splits of the WTCCC data into training and test sets, we

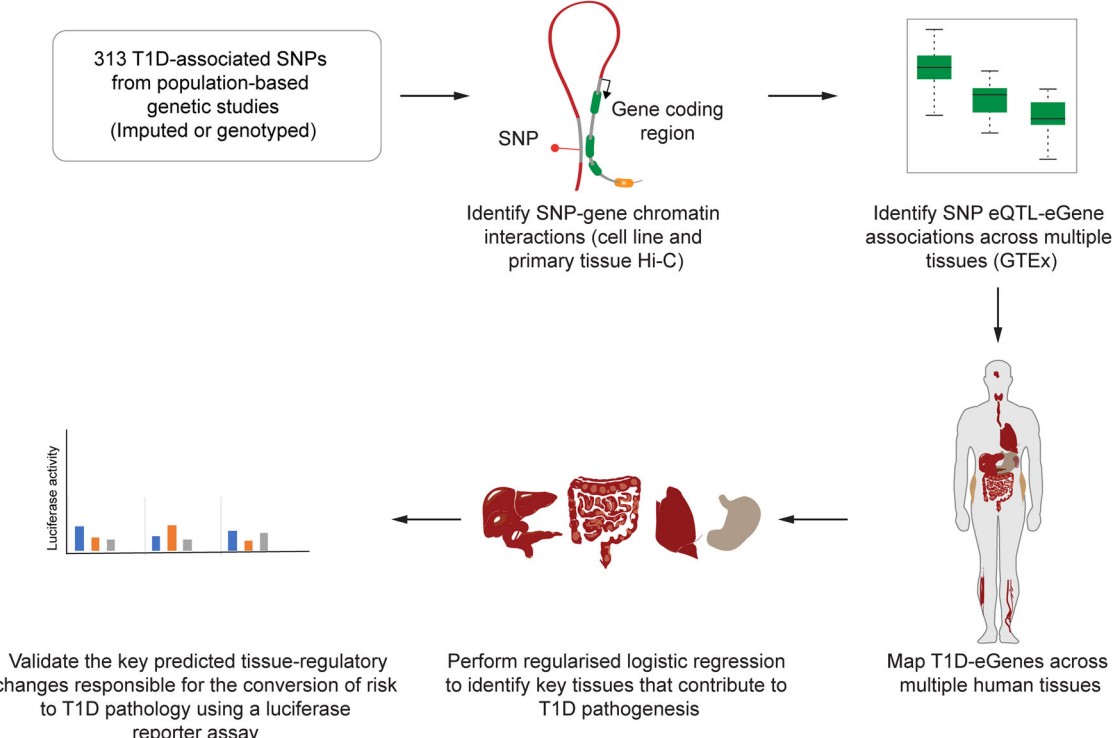

**Fig. 1 Overview of the method used to rank the tissue-specific transcription regulatory effects of genetic variants associated with the development of T1D.** SNPs associated with T1D were analysed (CoDeS3D) to identify the interacting loci, which were then tested for eQTLs within GTEx. This resulted in a map of T1D-eGenes across multiple human tissues. The T1D genes were subsequently used to perform logistic regression to identify key tissues for T1D pathogenesis. Luciferase assays were used to test the final predictions.

sampled 50 different training and test sets by repeating a 5-fold internal cross-validation of the WTCCC data a total of 10 times. This experiment generated 50 out-of-sample AUC values from 50 different T1D regularised logistic regression predictors created with model 1's optimised hyperparameters. Each predictor was trained on a different subset comprising 80% of the WTCCC data and evaluated on the remaining 20% (Fig. 2; Supplementary Data 7 and Methods section). The 50 out-of-sample AUCs varied between 0.712 and 0.771 with a mean of 0.747 and a standard deviation of 0.14 (Supplementary Fig. 3). Internal cross-validation is a standard practice in machine learning[9] and showed that model 1 predictor, which was fitted with optimised hyperparameters, generalises well across different subsets of WTCCC derived eQTL data.

Tissue-specific contributions to the T1D risk were extracted from each of the 50 T1D regularised logistic regression predictors as the sum of the absolute values of the model weights associated with each tissue. We then ranked the tissue-specific contributions to the 50 regularised logistic regression predictors. This ranking identified the lung as the top average contributor to the relative risk (case:control) of developing T1D. Across all 50 regularised logistic regression predictors, the lung explained a mean of 13.6% (standard deviation of 2.51%) of the relative risk of developing T1D (Fig. 2; Supplementary Data 8).

***CTLA4*** **contributes to the risk associated with the lung and testes.** When training our T1D regularised logistic regression predictors, we identified a split distribution for the lung that was dependent upon the lasso regularisation inclusion or exclusion of the rs3087243-*CTLA4* cis-acting (i.e. <1 Mb apart) spatial eQTL within the two tissues where it was identified (i.e. lung or testes; Supplementary Fig. 4). Since lasso regularisation retains only one of a group of highly correlated features, we sought to validate the

possibility that rs3087243-*CTLA4* has notable effects on the contribution of the lung and testis to T1D risk.

To test the effects of specifically removing the rs3087243-*CTLA4* feature, we created two alternative weighted WTCCC genotype T1D-eQTL models in which this eQTL was removed from either the lung or the testis within the weighted WTCCC genotype T1D-eQTL matrix. The AUCs and tissue-specific contributions from 50 T1D predictors with the optimised hyperparameters from each of the alterative matrices were evaluated by two-sided *t*-test and Bayesian methods (Supplementary Figs. 5 and 6; ref. [10]). The inclusion or exclusion of the rs3087243-*CTLA4* spatial eQTL within the lung or testes had a strong impact on the lungs' predicted contribution to T1D risk. No other tissues were affected, consistent with the rs3087243-*CTLA4* spatial eQTL only being detected in the lung and testes. The lung rs3087243-*CTLA4* eQTL contributed an average of 4% to T1D risk (Supplementary Data 9). The lung eQTL involving rs3087243 and *CTLA4* is also notable as: (1) rs3087243 has been linked with progression from single to multiple autoantibodies in the TrialNet PTP cohort[11]; (2) *CTLA4* encodes an immunoglobulin protein crucial for modulating T cell function and mediating autoimmunity and (3) immune intervention trials targeting *CTLA4* have reported significant but short-term positive metabolic outcomes[12].

**Predictions from the first model were confirmed using a second model with more features.** To enable more precise estimations of relative risk for data features, a second T1D regularised logistic regression predictor (model 2) (134 tissue-specific eQTL effects across GTEx tissues and six SNPs with unknown eQTLs) was created (Methods) and trained with the optimised hyperparameters using the full WTCCC cohort (in-sample AUC = 0.774). (Model 2 differed from model 1, which used an 80:20 split for internal WTCCC validation.) Model 2 was validated against the UK Biobank (UKBB)

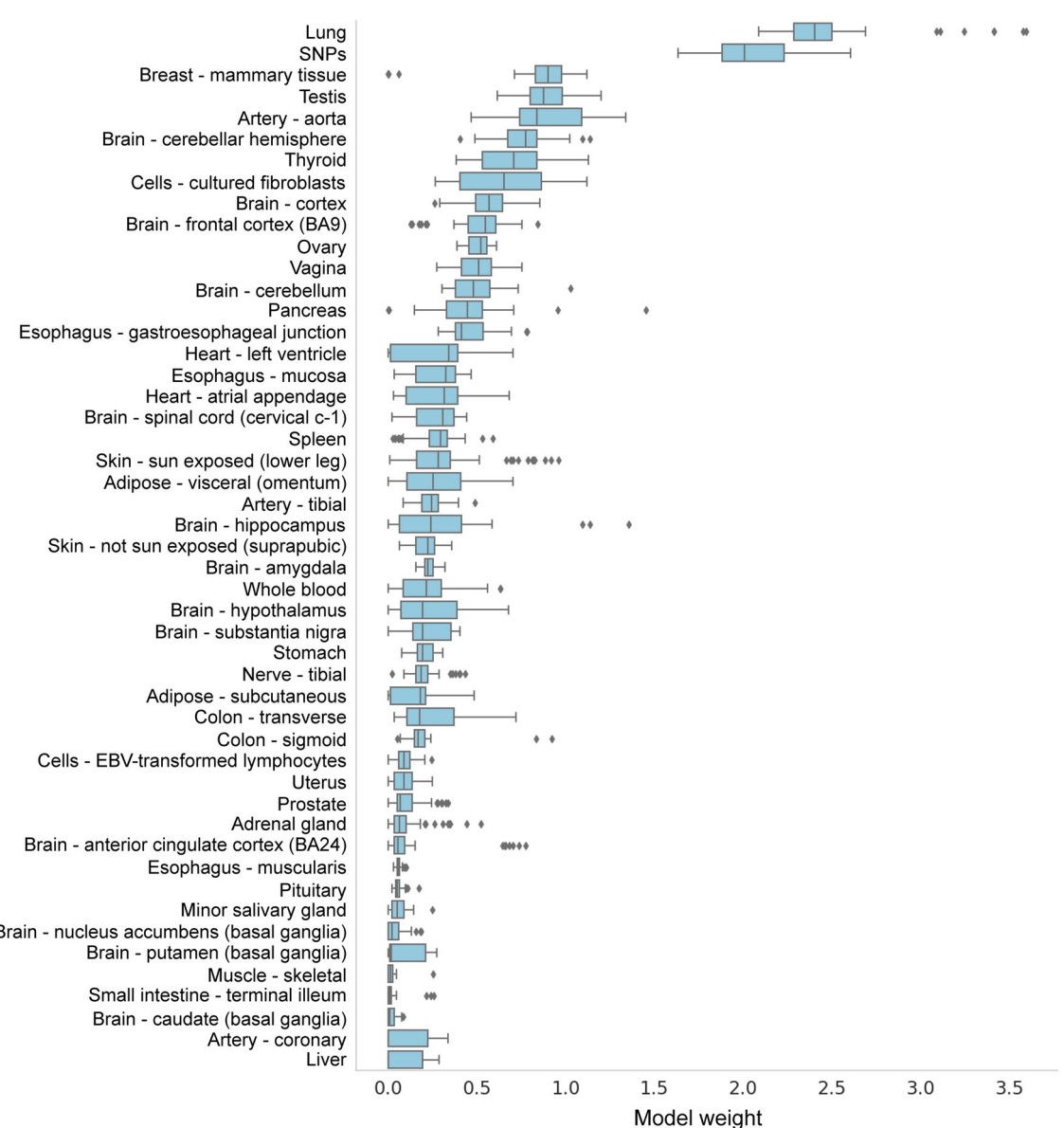

**Fig. 2 Tissue-specific contributions of the 50 T1D regularised logistic regression predictors.** The distributions were created from the 50 T1D regularised logistic regression predictors that were created using the optimised hyperparameters of model 1. These regularised logistic regression predictors integrated four different forms of biological information: GWAS or fine-mapping; Hi-C; eQTL and genotype data. SNPs denote T1D-associated SNPs for which no eQTLs were identified. The X axis is the total value of the model weights (no units).

(30 subsets of T1D datasets of 415 cases and 578 controls). Model 2 achieved a mean AUC = 0.754 (Table 1 and Fig. 3a [Kernel density estimate] and Fig. 3b [posterior estimation results from the Bayesian analysis]; Supplementary Data 10) and was used to rank the eQTLs that impacted the lung contribution to T1D risk (Supplementary Data 11). It should be noted that model 2 excluded the rs3087243-*CTLA4* eQTL, which contributed an average of 4% to T1D risk in the 50 T1D regularised logistic regression predictors (calculated with the optimised hyperparameters of model 1; see above). The major contributor eQTL (rs6679677) downregulated *AP4B1-AS1* transcript levels in cis (i.e. rs6679677 and *AP4B1-AS1* are <1 Mb apart) in the lungs and conferred a 13.3% contribution to the T1D regularised logistic regression predictor. This was comparable to the mean predictor weighting (13.6%) of the 50 models created with model 1's hyperparameters. Notably, rs6679677 also downregulates *AP4B1-AS1* expression in whole blood samples (eQTLGen; Supplementary Data 12) and modulates the expression of genes associated with

immune regulation, including *FOXP3*, *CTLA4*, *IL2RA* and *SLAMF1* in whole blood (eQTLGen; http://www.eqtlgen.org [13]; Supplementary Data 12).

**Table 1 A summary from the Bayesian analysis of the validation AUCs from the model 2 predictor on the 30 UK Biobank dataset.**

|          | Mean  | SD    | HPD_2.5% | HPD_97.5% |
|----------|-------|-------|----------|-----------|
| $\mu$    | 0.754 | 0.002 | 0.749    | 0.758     |
| $\sigma$ | 0.011 | 0.002 | 0.009    | 0.014     |

Uninformed prior distribution was normal with mean –U(0,1) and stand deviation –HalfNormal(std = 0.01).
$\mu$ mean of the simulations (1000) from the model created using 30 AUC prediction results, $\sigma$ standard deviation from the simulations, *Mean* simulated data mean, *SD* standard deviation, *HPD* highest posterior density interval, *AUC* area under the curve.

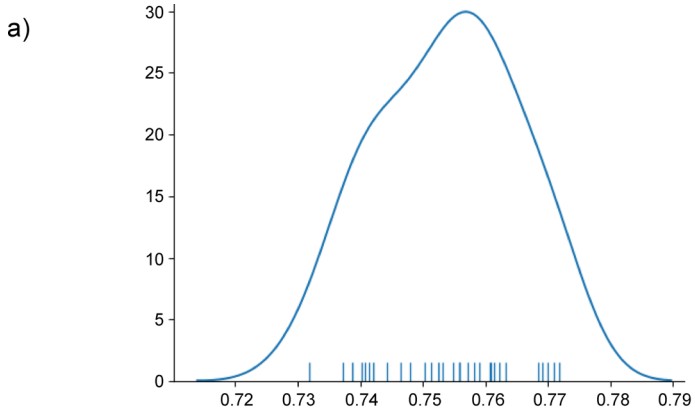

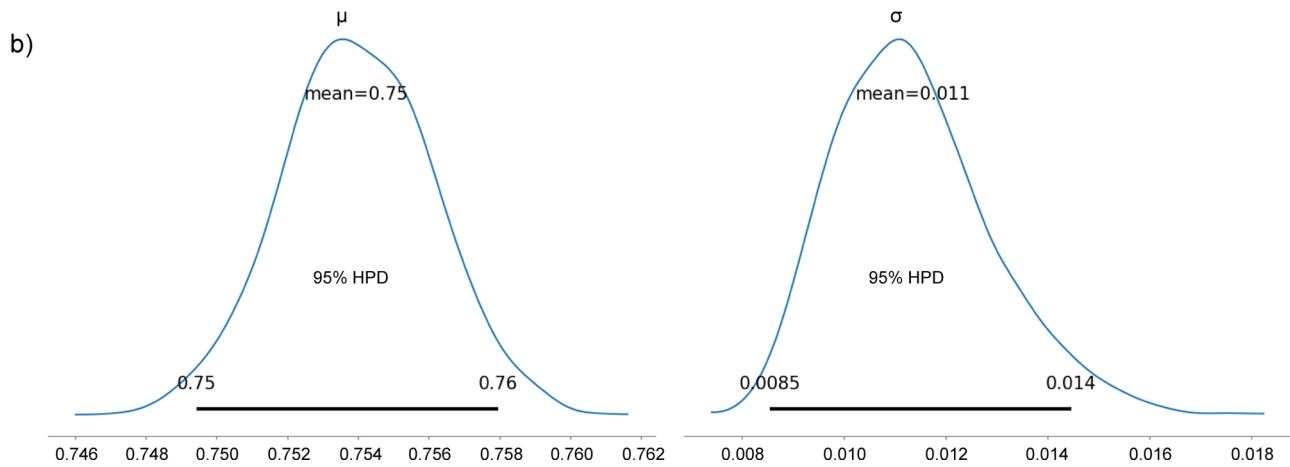

**Fig. 3 Validation of AUC results from Model 2 on the 30 UK Biobank test dataset. a** Kernel density estimate (KDE) plot of the 30 AUC results (mean = 0.754; SD = 0.0011. [X axis: AUC, Y axis: frequency of AUC values (%)]. **b** The posterior estimation results from the Bayesian analysis (1000 iterations of model simulation) of the 30 AUCs. From the Bayesian estimation, there was a 95% chance that the mean ($\mu$) AUC was between 0.749 and 0.758, and its SD ($\sigma$) was between 0.009 and 0.014. SD standard deviation, AUC area under the curve, HPD highest posterior density interval. Uninformed prior distribution was normal with mean ~ U(0,1) and standard deviation ~ HalfNormal(std = 0.01). [X axis: mean AUC and standard deviation of AUC, Y axis: posterior distribution].

**Regulatory changes in the HLA locus contribute to the risk of developing T1D in both models.** The HLA region is strongly associated with the development of T1D, accounting for 40–50% of the familial aggregation[14]. We observed spatial eQTLs involving HLA (Supplementary Data 13) within models 1 and 2. The notable HLA spatial cis-acting eQTLs involved SNPs rs2251396-*PSORS1C1*, rs2251396-*MICA*, rs3129889-*HLA-DRB5* and rs9268645-*HLA-DQB2* and were observed as contributing to the risk of developing T1D in multiple tissues (e.g. adrenal gland, transverse colon, small intestine, spleen, left heart ventricle, sun-exposed skin, testis and thyroid).

**Validation of lung cell allele-specific enhancer activity of locus marked by rs6679677.** Our results indicate that eQTL (rs6679677) downregulates *AP4B1-AS1* transcript levels in the lung. Therefore, we performed a luciferase enhancer assay to experimentally validate that the top-ranked eQTL (rs6679677) marks an allele and tissue-specific enhancer. DNA sequences flanking rs6679677 (i.e. 74 bp 5′ – ref/alt allele – 75 bp 3′ [chr1:114303734-114303884; GRCh37]) were cloned into the 3′ UTR of a minimal TATA-box promoter and luciferase gene construct to test whether the cloned sequence contain enhancer elements for gene expression (Methods section and Supplementary Fig. 7)[15]. Transient transfection of the plasmid vector

containing the reference locus (i.e. the major allele for rs6679677) resulted in a fold increase in luciferase activity when compared to the control vector in A549 (lung) and HepG2 (liver) cells (~11 and ~5 fold increase, respectively). This is consistent with the existence of H3K9ac histone modifications at the locus tagged by rs6679677 in both the lung and liver tissues (see HaploReg; https://pubs.broad institute.org/mammals/haploreg/haploreg.php). Notably, a significant allele-specific reduction in enhancer activity (i.e. nucleotide change from C > A) was observed only in the A549 cells ($p = 0.005$ [two-sided $t$-test]; Fig. 4), consistent with the identification of an eQTL involving this locus in the lung but not the liver. Collectively, these results support the allele-specific enhancer activity for the locus marked by rs6679677 in the lung.

**Discussion**

In the present study, we have used a logistic lasso regression model to integrate T1D case and control genotypes with spatial eQTL data to predict the relative tissue-specific contributions for the conversion of genetic risk to T1D. The regularised logistic regression models generated in this manuscript were validated by both internal cross-validation (model 1) which is standard practice in machine learning[9] and external cross-validation using the different cohorts (model 2). Notably, the discussion refers to conclusions drawn from model 2, with the most stringent

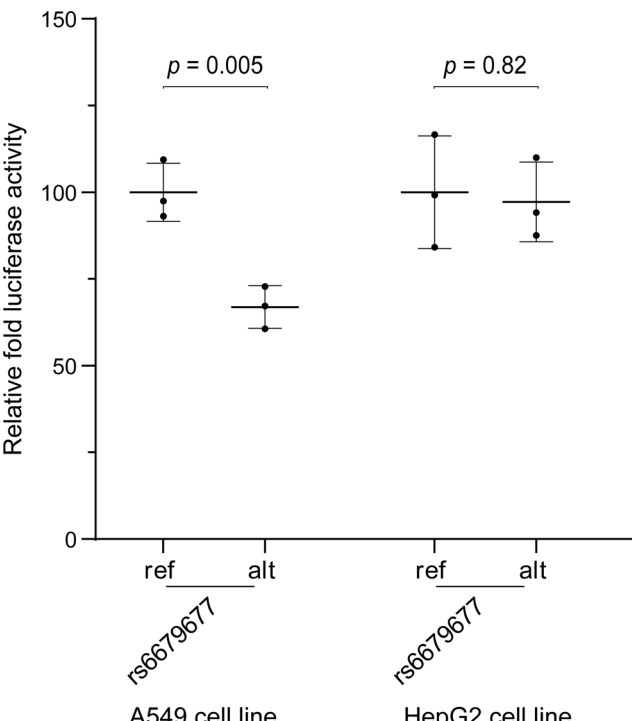

**Fig. 4 rs6679677 is an allele-specific enhancer (i.e. nucleotide change from C>A) in lung (A549) but not liver (HepG2) epithelial cells.** The locus tagged by rs6679677 was cloned within the 3′ UTR of a luciferase gene driven by a minimal promoter and transiently transfected into A549 and HepG2 cells. The relative enhancer activity for the ref and alt versions of the rs6679677 locus was calculated compared to the empty control vector (pMRAdonor2). Relative luminescence units (RLU) for the luciferase assay were normalised using the absorbance values for the beta-galactosidase assay. Results were plotted as the percentage of the ref_alleles enhancer activity (mean ± SD). Transfection experiments were repeated three times across $n = 3$ or $n = 9$ technical replicates in HepG2 (liver) and A549 (lung) cells, respectively. Representative results are shown for one transient transfection of A549 and HepG2 cells. Two-sided $t$-test was used for independent mean evaluation.

cross-cohort validation. Model 2 was validated across two independent case–control cohorts and identified the lungs and a cis-acting eQTL involving rs6679677 as the largest single contributor of the risk for developing T1D. Our results agree with observations by Gamazon et al.[16], who used LD score regression to identify the lung as the top tissue influencing T1D disease development. These observations are consistent with an environmental event impacting the largest single interface between the body and environment to precipitate the development of T1D[17–19]. Moreover, our results are consistent with the association of respiratory infections (i.e. influenza-like illness) with an increased risk of islet autoantibody seroconversion in young-onset T1D study cohorts[19,20]. Collectively, our results demonstrate that tissue-specific approaches can improve our understanding of disease aetiology, potentially aiding therapeutic development in preventing T1D onset.

It is widely recognised that the vast majority of measurable genetic risk for T1D is associated with the HLA region and polymorphisms of the class II HLA DQ, DR and DP genes[14]. Typically these effects are ascribed to polymorphisms within the HLA genes that affect the shape of the peptide-binding grove and the scope of the peptides that can bind to the allele and thus be presented on the cell surface[14]. Our analysis focused on

identifying and ranking the tissue-specific regulatory changes and thus does not capture variants that change the function or structural properties of a gene-encoded protein. Therefore, it is notable that the HLA locus variants that were retained within our models (rs3129889-*HLA-DRB5* and rs9268645-*HLA-DQB2*) regulated the expression of DRB and DQB alleles. In addition, rs2251396 was associated with the expression of *PSORS1C1*, which has been associated with T1D[21]. Similarly, rs2251396 was associated with the expression of *MICA*, which has previously been identified as part of an extended HLA haplotype that associates with TID risk[22], Collectively these observations support the hypothesis that changes in HLA gene expression contribute to T1D risk, in addition to the recognised role for HLA polymorphisms.

In the lung, our eQTL analysis revealed a marked association between rs6679677 and the expression of *AP4B1-AS1*. The allele-specific enhancer activity of the rs6679677 tagged region in lung cells was confirmed using a plasmid-based luciferase assay. However, rs6679677 also: (1) has eQTLs with *AP4B1-AS1* and immune regulation genes (i.e. *FOXP3*, *CTLA4*, *IL2RA* and *SLAMF1*[23–26]) in whole blood; (2) has been associated with the development of multiple persistent autoantibodies (including islet autoantibody), but not progression to T1D development in the TEDDY prospective cohort[27]; (3) has been associated with the development of other autoimmune disorders (i.e. juvenile idiopathic arthritis and rheumatoid arthritis)[28,29] and (4) was reported as the top non-HLA SNP associated with T1D from WTCCC studies[30]. Collectively, these results support an important molecular role for the locus tagged by rs6679677 in a lung-specific increase in the risk of T1D development. However, our results do not prove the effect is exclusively due to the impact on lung cells. As such, future work should dissect if the lung-specific regulatory impact of rs6679677 contributes to the mechanism of T1D risk.

*AP4B1-AS1* is located in a genomic region that is recognised as being strongly associated with autoimmune disorders[1]. *PTPN22*, which is on the antiparallel DNA strand to *AP4B1-AS1*, encodes a lymphoid-specific intracellular phosphatase (LyP) from the non-receptor class 4 subfamily of the protein-tyrosine phosphatases that acts as a critical negative regulator of T cell activation and T cell receptor signalling pathways[31]. Notably, rs6679677 down-regulates *AP4B1-AS1* expression in whole blood samples. Similarly, C1858T-rs2476601 (in complete linkage with rs6679677) has been linked with reduced protection against the influenza virus[32]. Therefore, we propose that future studies should ascertain the regulatory roles of rs6679677 on *AP4B1-AS1* in T cells, particularly in response to viral infections in the lungs. We contend that this will help untangle the genetic mechanisms that connect respiratory infections and the induction of islet autoantibodies that have been observed in young children[19,20].

This study has limitations. Firstly, the genetic data used in the analyses is predominantly from people of European ancestry, which limits the immediate translation of our findings to populations with different genetic structures (e.g. variable haplotypes in this region). Secondly, thirty SNPs identified as being associated with T1D did not have identifiable eQTLs in any of the GTEx tissues studied, consistent with the presence of alternative methods or developmental stages through which SNPs can mediate their effects on phenotypes. Thirdly, not all of the eQTLs we identified were represented in the individual genotypes we analysed (i.e. SNPs were unable to be imputed), meaning we could have missed effects. Fourthly, the reporter assay does not take into account the genomic context through which chromatin looping influences the enhancer-promoter interactions that mediate transcriptional activity[33]. Fifthly, the use of a lung cancer cell line may limit the interpretation of the transcriptional control

of genes in normal lung tissue. Lastly, most of the spatial chromatin interactions were identified from immortalized cancer cell lines or primary tissues. By contrast, the eQTL associations were primarily obtained from post-mortem samples taken from a cross-sectional cohort (20–70 years of age). Therefore, it is possible that the Hi-C interactions and eQTL sets were not representative of the tissues in which they were tested. Nevertheless, the reproducibility of the prediction model across independent cohorts supports the utility of the approach and its use with expanded datasets for T1D and other immune and non-immune diseases.

The novelty of the approach we undertook lies in: (1) the integration of T1D-associated SNPs with their tissue-specific eQTLs (in both cis and trans); (2) interpreting individual case and control genotypes in terms of these tissue-specific eQTL effects; (3) including effects from variants that do not have detectable eQTLs in the reference library that is used in the assay and (4) the application of machine learning to select and rank the tissue-specific eQTL effects that confer disease risk. This approach moves T1D research away from a candidate gene approach to include gene regulatory changes, including within the HLA locus, as possible contributors to the risk of developing T1D. However, all results are putative until they are followed up by integrative empirical methods that prove the link between gene expression in the lung and other tissues, and the conversion of T1D risk.

In conclusion, our work provides insights into the role of variation in gene regulation in the risk of developing T1D. The transcriptional changes (including *AP4B1-AS1* and *CTLA4*) we identified in the lung may help explain the reported association between respiratory infections and risk of islet autoantibody seroconversion reported in young children.

## Methods

**Identification of genetic variants associated with the development of T1D**. In total, 313 genotyped and imputed SNPs associated with T1D, and those associated with time-to-event development of islet autoimmunity and T1D, were retrieved from the GWAS catalogue (www.ebi.ac.uk/gwas, downloaded 8 February 2019; *p*-value $1 \times 10^{-5}$), prospective studies[27,34–36], TrialNet PTP cohort[11], adult-onset[37] and GRS prediction studies[38–40] (Supplementary Data 1). All genomic positions for SNPs and genes are annotated according to reference human genome hg19/GRCh37.

**Identification of SNP-gene pairs and expression QTL associations in human tissues**. The Contextualizing Developmental SNPs in three-dimensions algorithm (CoDeS3D[7]) was used to identify genes that physically interact with loci marked by the T1D-associated SNPs. Briefly, the CoDeS3D modular python scripts integrate Hi-C contact libraries from published sources (Supplementary Data 2) to identify spatial co-localisation of two DNA fragments, with one fragment marking the queried SNP. Gene-containing restricted fragments that are in physical contact with fragments containing the queried SNPs are identified as spatial pairs to the SNPs. Finally, the resultant spatial SNP-gene pairs are queried in the Genotype-Tissue Expression database (GTEx) to identify SNPs that are associated with transcript levels of genes through physical interaction at FDR < 0.05[7].

In the present study, spatial interactions were identified in Hi-C chromatin contact libraries captured from: (1) immortalized cell lines that represent the embryonic germ layers (i.e. HUVEC, NHEK, HeLa, HMEC, IMR90, KBM7 and K562); (2) B-cell derived lymphoblastoid cell line (GM12878) and (3) primary human tissues (i.e. spleen and pancreas) (Supplementary Data 2). These published libraries were chosen to ensure a wide range of known interactions and were analysed in aggregate within the pipeline to address a lack of tissue-specific Hi-C interactions captured within 'relevant' published Hi-C libraries. The identified putative spatial interactions aggregated from different Hi-C libraries were utilised to select tissue-specific eQTL effects. Of note, the summary statistics for the CoDeS3D run include the Hi-C cell line/tissue in which the interaction was observed (Supplementary Data 3; column = cell lines). The regulatory potential of the spatial connections was identified by incorporating eQTL information from 44 human tissues (Genotype-Tissue Expression database [GTEx] v7; www.gtexportal.org)[41]. Spatial eQTLs were deemed significant and recorded if the *q* < 0.05 after correcting for multiple testing using the BH procedure[42]. Finally, genes whose transcript levels were associated with a spatial-eQTL were denoted as eGenes. The eQTL-eGene interactions were defined as either: *cis*, the eQTL and eGene are separated by a linear distance of ≤1 Mb on the same chromosome; or *trans*, eQTLs and their eGenes were separated by >1 Mb on the same chromosome or located on different chromosomes.

**Genotype imputation for T1D cases and controls**. Genotypes from T1D cases (2000) and controls (3000) were obtained from the Wellcome Trust Case Control Consortium (WTCCC)[30]. SNPs within individual genotypes were converted to rsIDs and genomic positions mapped (GRCh37, hg19). PLINK (v1.90b6.2, 64-bit) was used for quality control. Genotypes were cleaned using the Method-of-moments F coefficient estimate to remove homozygosity outliers (*F* values < −0.04 or 0.025 < *F* values). Related individuals were identified and removed using proportion IBD (PI_HAT > 0.08). Ancestry outliers (identified by principal component analysis [PCA] plotting), individuals with sex genotype errors (identified by PLINK), or individuals with missing genotype data (missing rate > 5%) were also removed. Finally, SNPs that were not in Hardy–Weinberg Equilibrium (*p* < 10^{-6}) or had a minor allele frequency <1% were removed before SNP data imputation (Sanger imputation server; https://imputation.sanger.ac.uk)[43]. PLINK and bcftools (version: 1.9) were used in the imputation data preparation specified by Sanger imputation service[43]. Following imputation, the T1D genotype data was cleaned to remove SNPs with an: impute2 score <0.3; missing data rate >5% or a minor allele frequency <1%.

**Creation of a weighted WTCCC genotype T1D-eQTL matrix**. The machine learning only used SNPs (quantified or imputed) that did not have any missing data across the cohort and thus were present within each of the genotypes that were used. From the total 313 T1D SNPs included in the study, 253 SNPs were present within each of the WTCCC genotypes (Supplementary Data 6). Of these 253 T1D SNPs, 224 had detectable eQTLs, connecting to 758 eGenes (6307 tissue-specific eQTL effects). The tissue-specific eQTL normalised effect size for each T1D-associated SNP within the imputed WTCCC genotypes was extracted from the GTEx eQTL summary table of significant eQTLs (Supplementary Data 3). The normalised effect size for each tissue-specific eQTL was weighted by the number of alternative alleles at the eQTL SNP position in each individual's genome. The 30 T1D-associated SNPs that were not eQTLs were unweighted, using solely SNP allele count from the imputed genotype.

**Generation, training and validation of the regularised logistic regression models**. In order to identify the optimised predictor model parameters, the weighted WTCCC genotype T1D-eQTL matrix was randomly split (80:20) by python numpy-permutation into two groups that contained case and control genotype data for model training and validation. The Mann-Whitney U test[8] (tsfresh version 0.12.0[44]) was used to select the individual feature columns within the 80% training dataset that were the most relevant attributes for predicting the T1D status (i.e. the relevant subset; FDR = 0.2)[45]. The relevant subset was then used to train a multiple logistic algorithm (Scikit-learn version 0.21.3[46];) implemented with elastic net regularisation using the SAGA solver to predict T1D disease status. The training was optimised using a Grid Search algorithm with 10-fold cross-validation to identify the best predictor with the optimised parameters (model 1). Hence, model 1 was created from 80% of the data with the optimised parameters (hyperparameters: C = 1, l1_ratio = 1, max_iter = 500, penalty = 'elasticnet', random_state=1, solver = 'saga').

We used prediction performance (AUC) to enable us to identify the best performing models for subsequent use. Prediction performance (measured by area under the curve [AUC]) for model 1 was tested using the relevant subset from within the 20% validation dataset. The optimal hyperparameter l1_ratio=1 effectively reduces the elastic net regularisation to a lasso regularisation. We used Elastic net regularisation and found via hyperparameter optimisation that the limit case of lasso regularisation was the most performant.

To calculate a measure of the variation in AUCs of the modelling with the optimised parameters; we undertook ten repeats of 5-fold cross-validation of model generation and validation using the Scikit-learn RepeatedKFold algorithm[46], starting with the random generation of the 80:20 training:validation data sets and without Grid Search optimisation. This resulted in 50 T1D logistic regression predictors derived using the same general parameters as model 1.

**Calculation of tissue-specific contributions to T1D risk**. The 50 T1D regularised logistic regression predictors created from the 10 repeats of 5-fold cross-validation were used to test the predictive power of tissue-specific eQTL effects on individual genotype risk scores. Tissue-specific contributions to the T1D risk were extracted from each predictor as the sum of the absolute values of the weights associated with each tissue.

**Validation of the importance of the lung eQTLs in UK Biobank data (model 2)**. A second model (Supplementary Fig. 2) was created and trained using the full WTCCC training dataset with the optimised parameters. This model did not use the 80:20 split that was used in model 1. The predictor was then validated using 30 cohorts of 993 individual samples (415 cases and 578 controls) derived from the UK Biobank (Supplementary Data 9).

The 415 cases were selected, using a modification of Sharp et al.[39], from the UK Biobank imputed (487,411 individual samples) BGEN format dataset using the following criteria:

(1) European Caucasian by genetic clustering methods
(2) Clinical diagnosis of diabetes at ≤ 20 years of age
(3) On insulin within 1 year from the time of diagnosis
(4) Still on insulin at the time of recruitment
(5) Never self-report as having type 2 diabetes (T2D)
(6) All SNPs included in the model 2 predictor were present within each individual's imputed genotypes

The 578 control individual samples, without missing data for any of the SNPs included in the model 2 predictor, were randomly selected from the healthy controls within the UK Biobank data for each of the 30 test datasets. The genotype data for the 993 case and control UK Biobank samples in each test dataset was used to build a weighted eQTL-genotype matrix as outlined for model 1.

**Reporter assay methodology for validating the regulatory effects of genetic sequences.** Luciferase reporter assays (Supplementary Fig. 7) were performed using a modification of[15]. Briefly, DNA sequences flanking rs6679677 (i.e. 74 bp 5′-ref/alt allele – 75 bp 3′ [chr1:114303734-114303884; GRCh37]) containing the reference and alternative sequences) were synthesised by Integrated DNA Technologies (IDT). To ensure compatibility with the pMPRA vectors (pMPRA1 [Addgene: plasmid #49349] and pMPRAdonor2 [Addgene: plasmid #49353]), each sequence was designed using the following template: 5′-ACTGGCCGCTTCACTG-var-GGTACCTCTAGAAGATCGGAAGAGCGTCG-3′ (i.e. var denotes the 150 bp sequence to be assayed) (Supplementary Fig. 7). The variable region (var) was separated by a pair of *Kpn*I (GGTACC) and *Xba*I (TCTAGA) restriction sites to enable directional insertion of a reporter gene. PCR amplification was performed using primer sequences (MPRA_SfiI_F [forward], MPRA_SfiI_R [reverse]; Supplementary Data 13) to add two distinct *Sfi*I (GGCCNNNNNGGCC) tails to enable directional ligation of the oligonucleotide into pMPRA1 (Supplementary Fig. 7). An aliquot of the amplification product was electrophoresed (2% agarose, 100 V, 45 min) to visualise and verify that the product was the correct size (~200 bp) and that there were no non-specific amplification products. PCR amplicons were digested with *Sfi*I (50 °C, 2 h), purified (QIAquick PCR Purification Kit; Qiagen) and quantified by Nanodrop.

pMPRA1 was linearised by digesting with *Sfi*I (50 °C, 2 h), electrophoresed (0.8% agarose, 60 V, 1 h), the 2.5 kb linearised vector backbone excised, gel purified (Zymoclean™ Gel DNA Recovery Kit; Zymo Research) and quantified by Nanodrop.

*Sfi*I-digested oligonucleotides (100 ng) were mixed with linearised pMPRA1 vector backbone (50 ng) and ligated by T4 DNA ligase (1U, 16 °C, min) to create pMPRA1: rs6679677_ref and pMPRA1: rs6679677_alt. The ligation reaction was stopped by heating (65 °C, 20 min).

pMPRA1:rs6679677_ref and pMPRA1:rs6679677_alt were amplified and selected by transformation in competent *E. coli* DH5-alpha cells (Mix & Go competent cells) according to the manufacturer's instructions (www.zymoresearch.com/). Briefly, competent *E. coli* DH5-alpha cells (100 μL) were thawed on ice before the addition of ligation products (1-5 μL), gentle mixing (by flicking) and incubation on ice (5 min). Immediately following incubation on ice, the transformed competent cells (100 μL) were spread onto pre-warmed LB agar plates supplemented with ampicillin (100 μg/mL) and incubated (37 °C, overnight). Single colonies were picked and inoculated into LB:Ampicillin media (5 ml containing 100 μg/mL Ampicillin) and incubated (37 °C, overnight) with shaking (~200 rpm). Plasmid DNA was extracted using a QIAprep Spin Miniprep Kit, according to the manufacturer's instructions. Plasmids were Sanger sequenced (Massey Genome Service; Massey University) using RVprimer3 (forward) and EBV-rev (reverse) primers in (Supplementary Data 13) to confirm the sequences of the inserts.

pMPRA1:rs6679677_ref, pMPRA1:rs6679677_alt were linearised with *Kpn*I (10 U, 37 °C, 1 h) and purified using the QIAquick PCR Purification Kit (Qiagen). Samples were subsequently digested with *Xba*I (10 U) in the presence of Shrimp Alkaline Phosphatase (1 U, 37 °C, 2 h) prior to heat-inactivation (65 °C, 5 min) and purification using the QIAquick PCR Purification Kit (according to the manufacturer's instructions).

A luc2 open reading frame (ORF) was prepared from pMPRAdonor2 (1 μg) by *Kpn*I (20 U) *Xba*I (20 U) double digestion (37 °C, 1 h), electrophoresis (0.8% agarose, 60 V, 1 h) and gel purification of the 1.7 kb band using the Zymoclean™ Gel DNA Recovery Kit (Zymo Research).

The *luc2* open reading frame was cloned into the *Kpn*I and *Xba*I digested pMPRA1:rs6679677_ref, pMPRA1:rs6679677_alt plasmids, transformed and selected as described earlier. The resultant plasmids (pMPRA1:luc_rs6679677_ref, pMPRA1:luc_rs6679677_alt) were Sanger sequenced (Massey Genome Service; Massey University) using the luciferase primer in (Supplementary Data 14) to confirm the *luc2* gene insertion.

HepG2 and A549 cell lines were purchased directly from the American Type Culture Collection (ATCC) and used at an early passage number. All cell lines tested negative for mycoplasma contamination and no commonly misidentified cell lines were used in the study. A549 (lung epithelial carcinoma; ATCC) and HepG2 (human liver carcinoma; ATCC) cells were maintained in DMEM and RPMI 1640 (ThermoFisher), respectively, supplemented with 10% fetal bovine serum, 1% GlutaMAX and 1% penicillin/streptomycin at 37 °C in a humid incubator purged

with 5% $CO_2$. Cells were routinely tested for mycoplasma contamination. For transfection, ~$1.0 \times 10^5$ cells were seeded in a single well of a 24-well plate, followed by the addition of 500 μL of the appropriate complete media. On the day of transfection (24 following cell plating), ~75% confluent wells were co-transfected with luciferase plasmid DNA (i.e. 800 ng of pMPRA1:luc_rs6679677_ref, pMPRA1:luc_rs6679677_alt, or pMPRAdonor2 luciferase control) and a beta-galactosidase control plasmid (200 ng) using lipofectamine 3000 (ThermoFisher; according to the manufacturer's instructions).

At 48 h following transfection, cells were lysed using the Glo lysis buffer (Promega) and luciferase activity assessed using the ONE-Glo™ Luciferase Assay System (Promega) in a VarioskanTM LUX multimode microplate reader (according to the manufacturer's instructions). For the beta-galactosidase assay, 20 μl beta-galactosidase reagent (i.e. 0.2 M phosphate buffer (pH 7.4), 2 mM MgCl2, 100 mM β-mercaptoethanol and 1.3 mg/ml ortho-Nitrophenyl-β-galactoside) was added to 20 μl of transfection cell lysate (prepared using the Glo lysis buffer) in a 96 well plate and incubated at 37 °C for 30 minutes. The absorbance was then read at 420 nM in a VarioskanTM LUX multimode microplate reader following the manufacturer's instructions.

**Statistics and reproducibility.** All statistical testing was performed using R software (version v3.6.3)[47], Scikit-learn (version 0.21.3[46]), tsfresh (version 0.12.0[44]) and pymc3 (version 3.8[48];). Visualisation for the luciferase luminescence was performed using GraphPad Prism (v8.4.3).

Data security and code management is the foundation of reproducible and reliable data analyses. Datasets that were received or downloaded from original sources were individually maintained in read-only and write-protected directories on secured cloud server. Programme code was preserved after producing validated results and the code was named with appropriate functional and step sequential information. Version control (git) was also employed to protect script integrity across the analysis step directories.

**Reporting summary.** Further information on research design is available in the Nature Research Reporting Summary linked to this article.

## Data availability
All datasets generated for this study are included in the article as Supplementary Data. The Supplementary Data and the source data of Fig. 4 are available in figshare with the identifier [doi: 10.17608/k6.auckland.15071226][49]. This study makes use of data generated by the Wellcome Trust Case-Control Consortium[30], The Genotype-Tissue Expression (GTEx) Project and the UK Biobank Resource (Application Number 51306). The genotype data from Wellcome Trust Case and Control Consortium and UK Biobank are restricted to share by the data sharing agreements. Please refer to the organisations to obtain the genotype data. All other data related to this study are available from the corresponding author on reasonable request.

## Code availability
CoDeS3D pipeline is available at: https://github.com/Genome3d/codes3d-v1. Python scripts used for machine learning are available at: https://github.com/Genome3d/T1D_logistic_lasso_predictor.git/ and Zenodo with the identifier [doi: 10.5281/zenodo.5152705][50]. Python version 3.7.3 was used for all the python scripts. PLINK (v1.90b6.2, 64-bit) and bcftools version: 1.9 were used for SNP data cleaning and processing.

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

## Acknowledgements

J.M.O. is the guarantor for this article and takes full responsibility for the work as a whole, including the study design, access to data and the decision to submit and publish the manuscript. This research was supported by using the NeCTAR Research Cloud, a collaborative Australian research platform supported by the National Collaborative Research Infrastructure Strategy. Funding for the Wellcome Trust Case-Control Consortium was provided by the Wellcome Trust under award 076113 and 085475. The Genotype-Tissue Expression (GTEx) Project was supported by the Common Fund of the Office of the Director of the National Institutes of Health and NCI, NHGRI, NHLBI, NIDA, NIMH and NINDS. The authors wish to acknowledge the use of New Zealand eScience Infrastructure (NeSI) high-performance computing facilities, consulting support and/or training services as part of this research. New Zealand's national facilities are provided by NeSI and funded jointly by NeSI's collaborator institutions and through the Ministry of Business, Innovation & Employment's Research Infrastructure programme. URL https://www.nesi.org.nz. D.M.N. was funded by the Sir Colin Giltrap Liggins Institute Scholarship. D.H. was funded by grant UOAX1611: New Zealand-Australia Lifecourse Collaboration on Genes, Environment, Nutrition and Obesity (GENO) from the Ministry of Business, Innovation and Employment of New Zealand to JOS.

## Author contributions

D.M.N. ran the CoDeS3D and luciferase analyses. D.H. ran the machine learning analyses. D.H. and D.M.N. interpreted data and wrote the first draft. W.S., R.S., A.W.K.L. and J.M.O. co-supervise D.H. participated in discussions and commented on the manuscript. M.H.V., J.K.P. and J.M.O. co-supervise D.M.N., participated in discussions and commented on the manuscript. J.A.T. supervised D.M.N. performing lung cell enhancer assays and commented on the manuscript. J.M.O. was responsible for the overall direction of the study.

## Competing interests

The authors declare no competing interests.
