## [Transparent Peer Review File · Communications Biology]

Reviewers' comments:

Reviewer #1 (Remarks to the Author):

Overall comments:-

The title of this paper is :

"Machine learning identifies Regulatory changes in the lung as key to conversion of genetic risk to type 1 diabetes"

However I found it very hard to believe this message by the end of the manuscript. This study takes advantage of genetic associations and then uses integrated spatial genomics (using eQTL data) to characterise the functional impact of single SNPs. It uses world leading datasets of genetic associations and then interrogates their role in different tissues and at different times using information from these studies in the GTEX and DICE datasets. Their main result in their title and abstract is the expression of CTLA4 and AP4B1-AS1 in the lung is a major contributor to the conversion of T1D risk to the development of T1D.

A strength of this study is that there is significant potential in taking robust genetic associations of disease and utilising computational methods to interrogate their role in biology. Datasets such as GTEX and DICE offer a fantastic platform to try and drill down into the biology that is linked to these genetic associations. I have a concern that the novel approaches they have taken have not been thought through enough in relation to what is already known about biology, and whether they prove a mechanistic pathway is important, or simply raise it as a hypothesis.

the immune system is such a complex network of interactions at sub-cellular, cellular and organism level, I am sometimes left wondering whether they have 1. shown t1d associated loci have an impact on gene expression and immune regulation in specific tissues at specific times, or 2. that these associations are mechanistically important to conversion of autoimmunity, risk of t1d, or age of diagnosis. Their title and abstract suggest the second but as currently written I am struggling to find the evidence presented in the paper as persuading me that number 2 is true.

Specific comments:

It is reassuring that the associations they find in paragraph 1 and 2 of the results support existing literature and mechanistic studies. I think it is a valid contribution to try to dive into the detail of known genetic loci and their associations with cis and trans eQTL data, as they have done in the first two paragraphs of their results. I think their findings fit with and build on existing literature.

Paragraph 3 starting line 121- age association analyses. I got a bit more lost by their age association analysis. The top associated loci for t1d risk are SNPs in linkage with HLA DR3-DQ2 and HLA DR4-DQ8. These also have an association with age of diagnosis (although the association with age of diagnosis is weaker than the association with risk overall). This whole region has a complex linkage pattern. This means that it is hard to know if any individual associated SNP is having an effect itself or an effect by being a proxy for a linked allele. I could not understand how they disentangled the impact of a SNP on expression, versus a change in the HLA allele associated with the SNP having an impact directly on gene expression. Maybe I am missing something here but I really felt that this was not clear.

the genetic associations of age of diagnosis and seroconversion in t1d are modest, compared to the very strong associations of certain key genes with overall risk. I think it is valid to use existing gene expression datasets to suggest functions of associated genes, but I feel this is more hypothesis

generating rather than proof of a particular mechanism being relevant to t1d, and should come with a health warning.

Paragraph starting line 178:- in some ways maybe the most important part of the paper as it relates to the title and the main findings in the abstract. Machine learning identifies the lung as a susceptible site for conversion of risk to t1d:-

I think the investigators here have shown that certain groups of t1d specific genes have a strong impact on gene expression in specific tissues or cell types, and then related this back to the relative contribution of these tissues and gene expression changes to t1d risk. I am not convinced by the quantification of tissue specific contribution in this way. In my own mind a strong genetic association may have a strong impact on type 1 diabetes risk and may have a strong tissue specific effect, but the mechanism for the t1d risk may not be mediated via the strong tissue specific effect. It is possible that I have missed how they tested whether the tissue specific effect on gene expression impacted t1d risk this in my reading; but if this is the case I would strongly encourage the authors to present this more clearly.

the authors highlight that initiation of autoimmunity (which peaks at age 9 months in the TEDDY study) occurs at a different time to the study of gene expression in GTEx (average age of samples 60 years right?) and this really makes me wonder if the tissue specific contributions here can be extrapolated back to their impact on disease risk.

The authors have used ROC AUC, a measure of discriminative power (Janssens IJE 2020), to assess relative contribution of tissue specific eQTLs on t1d disease risk. If the goal here is improved prediction then their models as described perform less well than use of t1d associated genes alone, I don't think this is their goal. If the goal here is assessment of the relative contribution of different tissue specific effects on t1d risk then my main concern is that the pathways they investigate may have a strong impact on a certain tissue and t1d risk and I think the investigators show this, but I don't know this study shows that the impact of the genetic association is mediated via this particular tissue or cell type simply that it impacts both t1d risk and the cell type or tissue in DICE or GTEx. I am not sure, given ROC AUC is commonly used to assess predictive performance of machine learning models or diagnostic tests, that this is the best metric of information added by their modelling strategy, partly because this is so commonly used to assess predictive and diagnostic models, that the relatively worse performance compared to using genetic data alone at least needs to be explained or put in context to not leave the reader thinking "is there any information added from the gene expression level analysis"

My final comment:

I feel the manuscript is more hypothesis generating in its current form, the title and abstract may do better to reflect this. It may be that my comments above and my conclusion highlight that I have failed to understand their methods and results interpretation, if this is the case I think there is a good argument to simplify the paper and the way their study is presented to allow a wider audience to engage with the study.

Reviewer #2 (Remarks to the Author):

Nyaga et al. present a post-GWAS investigation of genetic variants identified in relation to Type 1 Diabetes (T1D). The current study builds on prior work by the same team of investigators in which they leveraged spatial organization of the genome by Hi-C to identify the cis- and trans-eQTL consequences of 180 T1D associated genetic variants. In the current investigation, the authors expand on their prior work by incorporating genetic variants from additional studies examining age-of-onset

for T1D, among others, yielding a total of 313 T1D related GWAS variants. After constructing the gene regulatory network of these 313 genetic variants, incorporating spatial and tissue specific eQTL information, the authors further apply machine learning approaches to build a predictive model to report the risk of conversion from T1D genetic risk to disease.

The manuscript incorporates multiple apparently state of the art techniques to address interesting biological research questions. My main concerns with the manuscript are the following: (1) the manuscript lacks a clear central message such that I am not sure precisely what I have learned even after reading it several times, (2) the manuscript glosses over critical details for many steps of analyses, making it difficult to assess whether the stated findings are technically sound, and (3) many of the findings of the manuscript relating to eQTL discovery and tissue specificity or proportion of variance attributable to particular tissues are not tied to clear and well-described statistical methods. These concerns, combined with the detailed points indicated below, make me hesitant to accept the major findings of this manuscript,

Detailed points:

1. Abstract: You write "we demonstrate that T1D SNPs are both tissue- and time of development-specific..." Do you mean that the T1D SNPs are tissue and/or time specific, or that the effects of these SNPs are tissue and/or time specific?
2. You emphasize the role of the lungs in your abstract and several other parts of the manuscript. However, your identification of overlap with eQTL in lung-tissue does not appear to provide conclusive evidence that T1D risk is tied to lung. GTEx has demonstrated that many eQTL are shared across tissues and, more recently, single cell RNA-seq studies have shown that specific cell populations can be shared across tissues. Is it possible that your identified "lung-specific" effects can be explained by representation of specific immune cell subsets in lung that are also relevant in other tissues/
3. Introduction: Multiple apparently factual literature-based statements appear to lack references. For example, the statement "Population level data has enabled genome-wide association studies (GWAS) that have identified ~60 risk regions that are associated with the risk of developing T1D" is unreferenced and should be supported by citation of prior T1D GWAS studies.
4. Introduction: the statement "The logistic lasso regression analysis of T1D case and control genotypes identified the lung, CTLA4 and AP4B1-AS1 genes as the greatest individual contributors to the conversion of the genetic risk for the development of T1D" is confusing. Do you mean that your prediction model identified importance for lung tissue overall, or for expression of specific genes are measured in lung?
5. Results, p. 4: You refer to your "in-house computational pipeline (CoDeS3D)..." Please clarify whether the in-house pipeline is the same or different from the publicly available github page. If different, please clarify what is different. In addition, please clarify in the Methods or otherwise what sources of data were used as input to the CoDeS3D pipeline and how.
6. Results, p. 4: Please introduce the major data sets used for analyses so that readers can better understand what resources were leveraged for the presented analyses. For example, when you mention "313 genotyped and imputed T1D-associated SNPs" for the first time, you should first clarify the cohort from which you obtained genotype data. I suggest you begin the results section with descriptives on the main cohort(s) used for statistical analyses, including the cohort names, sample sizes, age range, race/ethnic distribution, and other key characteristics.
7. Please clarify the methods used to call spatial eQTL and how these eQTL were assigned to genes, including the statistical significance thresholds used. In addition, I suggest you invoke some sort of eQTL colocalization and/or fine mapping approach in order to determine whether or not the T1D GWAS SNPs identified to be eQTL actually demonstrate causal relationships with the genes of interest.
8. Please clarify the set of features and search space considered in identification of the prediction model using lasso regularization.
9. The concept of "spatial genomics" is relatively new. Thus, the authors should clarify their meaning in using this term as well as "spatial eQTL" throughout the manuscript.

Re: COMMSBIO-20-2098A-Z

Dear Reviewers

We have re-written the manuscript to focus on the machine learning aspects. This has meant we have removed a large amount of additional material that cluttered the story line. We believe that this has significantly improve the manuscript.

Secondly, we were concerned about the strength of the data that was supporting the conclusions we drew. As such, we have extended the manuscript to include data from reporter assays that unequivocally support the fact that the region marked by rs6679677 is an allele specific enhancer within lung cells.

In so doing we believe we have fully addressed all concerns.

Sincerely

Justin M. O'Sullivan

Reviewer #1 (Remarks to the Author):

I am sometimes left wondering whether they have 1. shown t1d associated loci have an impact on gene expression and immune regulation in specific tissues at specific times, or 2. that these associations are mechanistically important to conversion of autoimmunity, risk of t1d, or age of diagnosis. Their title and abstract suggest the second but as currently written I am struggling to find the evidence presented in the paper as persuading me that number 2 is true.

- We thank the referee for their comments. We agree the initial manuscript was diffuse and poorly focused. We have now simplified the manuscript to remove discussion of age of onset/diagnosis and now concentrate solely on the machine learning to identify and rank the tissue specific contributions of the T1D-associated variants. The title has been changed to:

“Machine learning identifies the lung as a susceptible site for allele specific regulatory changes associated with risk for type 1 diabetes”.

We contend that this title is supported by the data we present, both computational and the new evidence of the allele-specific enhancer activity in the lung.

Specific comments:

Paragraph 3 starting line 121- age association analyses. I got a bit more lost by their age association analysis.

- We have removed this material from the manuscript.

I think the investigators here have shown that certain groups of t1d specific genes have a strong impact on gene expression in specific tissues or cell types, and then related this back to the relative contribution of these tissues and gene expression changes to t1d risk. I am not convinced by the quantification of tissue specific contribution in this way. In my own mind a strong genetic association may have a strong impact on type 1 diabetes risk and may have a strong tissue specific effect, but the mechanism for the t1d risk may not be mediated via the strong tissue specific effect.

- The referee is correct that one can envisage situations where a strong tissue-specific effect is not associated with the actual mechanism for the T1D risk. While this possibility cannot be ruled out by the results of our study, neither can the reverse. We contend that the experimental evidence of the allele-specific enhancer activity in the lung and the published evidence we cite support the hypothesis we present. However, it remains a hypothesis until tested in models or prospective cohorts. We believe we are clear on this point through-out the manuscript stating:

“However, our results do not prove the effect is exclusively due to the impact in lung cells. As such, future work should dissect if the lung-specific regulatory impact of rs6679677 actually contributes to the mechanism of T1D risk.”

The manuscript now ends with the statement:

“which could potentially explain the reported association between respiratory infections and risk of islet autoantibody seroconversion reported in young children.”

The authors have used ROC AUC, a measure of discriminative power (Janssens IJE 2020), to assess relative contribution of tissue specific eQTLs on t1d disease risk. If the goal here is improved prediction then their models as described perform less well than use of t1d associated genes alone, I don't think this is their goal. If the goal here is assessment of the relative contribution of different tissue specific effects on t1d risk then my main concern is that the pathways they investigate may have a strong impact on a certain tissue and t1d risk and I think the investigators show this, but I don't know this study shows that the impact of the genetic association is mediated via this particular tissue or cell type simply that it impacts both t1d risk and the cell type or tissue in DICE or GTEX. I am not sure, given ROC AUC is commonly used to assess predictive performance of machine learning models or diagnostic tests, that this is the best metric of information added by their modelling strategy, partly because this is so commonly used to assess predictive and diagnostic models, that the relatively worse performance compared to using genetic data alone at least needs to be explained or put in context to not leave the reader thinking “is there any information added from the gene expression level analysis”

- We thank the reviewer for their comment. AUC is only used as a measure of prediction performance to enable us to identify the best performing models in this study. We then used the good model to recognize the key body tissue types that contribute to T1D risk. We have included the following in the methods section:

“We used prediction performance (AUC) to enable us to identify the best performing models for subsequent use.”

I feel the manuscript is more hypothesis generating in its current form, the title and abstract may do better to reflect this. It may be that my comments above and my conclusion highlight that I have failed to understand their methods and results interpretation, if this is the case I think there is a good argument to simplify the paper and the way their study is presented to allow a wider audience to engage with the study.

- We agree the paper is hypothesis generating. We have now included empirical data to confirm the allele specific activity of rs6679677, have simplified the paper and changed the title. The title is now: “Machine learning identifies the lung as a susceptible site for allele specific regulatory changes associated with risk for type 1 diabetes”.

Reviewer #2 (Remarks to the Author):

(1) the manuscript lacks a clear central message such that I am not sure precisely what I have learned even after reading it several times.

- As stated above, we have revised the manuscript significantly to aid the readability and the clarity of the delivery of the main point.

(2) the manuscript glosses over critical details for many steps of analyses, making it difficult to assess whether the stated findings are technically sound.

- We have changed the manuscript to increase the accessibility of the critical details of the analyses. We have also provided access to github repositories of the code to ensure reproducibility.

(3) many of the findings of the manuscript relating to eQTL discovery and tissue specificity or proportion of variance attributable to particular tissues are not tied to clear and well-described statistical methods.

- We have improved the definitions of the models that were used to generate and attribute variance to tissues in the revised manuscript. We have further clarified the empirical methods. All the essential datasets and scripts are also provided in the supplementary material and github repository. Thus, we are confident that the work should be reproducible.

Detailed points:

1. Abstract: You write “we demonstrate that T1D SNPs are both tissue- and time of development-specific...” Do you mean that the T1D SNPs are tissue and/or time specific, or that the effects of these SNPs are tissue and/or time specific?

- We have re-written and simplified the abstract.

2. You emphasize the role of the lungs in your abstract and several other parts of the manuscript. However, your identification of overlap with eQTL in lung-tissue does not appear to provide conclusive evidence that T1D risk is tied to lung. GTEx has demonstrated that many eQTL are shared across tissues and, more recently, single cell RNA-seq studies have shown that specific cell populations can be shared across tissues. Is it possible that your

identified “lung-specific” effects can be explained by representation of specific immune cell subsets in lung that are also relevant in other tissues.

- The referee is correct that there is a possibility that the “lung-specific” effects could be explained by immune cell infiltration. In the manuscript we state that:

“Notably, rs6679677 also down-regulates *AP4B1-AS1* expression in whole blood samples (eQTLGen; Supplementary Table 12) as well as modulating the expression of genes associated with immune regulation, including *FOXP3*, *CTLA4*, *IL2RA*, and *SLAMF1* in whole blood (eQTLGen; <http://www.eqtlgen.org>¹⁶; Supplementary Table 12). “

- Moreover, we also state:

“However, rs6679677 also: 1) has eQTLs with *AP4B1-AS1* and immune regulation genes (i.e. *FOXP3*, *CTLA4*, *IL2RA*, and *SLAMF1*²³⁻²⁶) in whole blood; 2) has been associated with the development of multiple persistent autoantibodies (including islet autoantibody), but not progression to T1D development in the TEDDY prospective cohort²⁷; 3) has been associated with development of other autoimmune disorders (i.e. juvenile idiopathic arthritis and rheumatoid arthritis)^{28,29}; and 4) was reported as the top non-HLA SNP associated with T1D from WTCCC studies³⁰. Collectively, these results support an important molecular role for the locus tagged by rs6679677 in a lung-specific increase in risk of the development of T1D. However, our results do not prove the effect is exclusively due to the impact in lung cells. As such, future work should dissect if the lung-specific regulatory impact of rs6679677 actually contributes to the mechanism of T1D risk.”

3. Introduction: Multiple apparently factual literature-based statements appear to lack references. For example, the statement “Population level data has enabled genome-wide association studies (GWAS) that have identified ~60 risk regions that are associated with the risk of developing T1D” is unreferenced and should be supported by citation of prior T1D GWAS studies.

- We apologise for the omission of this citation. We have corrected this.

4. Introduction: the statement “The logistic lasso regression analysis of T1D case and control genotypes identified the lung, *CTLA4* and *AP4B1-AS1* genes as the greatest individual contributors to the conversion of the genetic risk for the development of T1D” is confusing. Do you mean that your prediction model identified importance for lung tissue overall, or for expression of specific genes are measured in lung?

- We have re-written the introduction. The final paragraph now reads:

“In the present study, we assigned SNPs associated with T1D to the genes they modulate through Hi-C chromatin interactions captured from primary tissues (*i.e.* pancreas and spleen) and immortalized cells. We integrated a regularised logistic regression model on European ancestry genotypes of T1D case and control to identify transcriptional changes in the lung involving *AP4B1-AS1* and *CTLA4* (associated with rs6679677) as the greatest individual contributors to the conversion of the genetic risk for the development of T1D. Finally, a plasmid-based luciferase reporter expression assay was performed to validate the allele specific enhancer activity of the locus marked by rs6679677 in lung cells.”

5. Results, p. 4: You refer to your “in-house computational pipeline (CoDeS3D)...” Please clarify whether the in-house pipeline is the same or different from the publicly available github page. If different, please clarify what is different. In addition, please clarify in the

Methods or otherwise what sources of data were used as input to the CoDeS3D pipeline and how.

- We apologise for the confusion. We have corrected this. CoDeS3D is correctly referenced and the github repository is cited.

6. Results, p. 4: Please introduce the major data sets used for analyses so that readers can better understand what resources were leveraged for the presented analyses. For example, when you mention “313 genotyped and imputed T1D-associated SNPs” for the first time, you should first clarify the cohort from which you obtained genotype data. I suggest you begin the results section with descriptives on the main cohort(s) used for statistical analyses, including the cohort names, sample sizes, age range, race/ethnic distribution, and other key characteristics.

- We have not presented the cohort characteristics for the studies that were used to identify the SNPs, or genotypes we used in this study. Rather we cite the studies in our manuscript. The SNPs were obtained from the GWAS catalog on the 8th of February 2019. We have cited this in the text and the provided the full list in Supplementary table S1.

7. Please clarify the methods used to call spatial eQTL and how these eQTL were assigned to genes, including the statistical significance thresholds used. In addition, I suggest you invoke some sort of eQTL colocalization and/or fine mapping approach in order to determine whether or not the T1D GWAS SNPs identified to be eQTL actually demonstrate causal relationships with the genes of interest.

- We have outlined the method for CoDeS3D in the methods section. In the results we state that the FDR of $q < 0.05$ was used when calling spatial eQTLs. The complete CoDeS3D output is presented in Supplementary Table S3.

We agree that the application of a colocalization analysis step would further reduce the inclusion of potential false positives in the machine learning steps. This is something we will do in the future.

8. Please clarify the set of features and search space considered in identification of the prediction model using lasso regularization.

- We have added additional points to clarify these parameters throughout the revised manuscript.

9. The concept of “spatial genomics” is relatively new. Thus, the authors should clarify their meaning in using this term as well as “spatial eQTL” throughout the manuscript.

- We have inserted the following definition upon first mention.

“We define a spatial eQTL as an eQTL in which the expression quantitative trait is associated with a SNP and gene that are captured as physically interacting within the nucleus”

Reviewers' comments:

Reviewer #1 (Remarks to the Author):

Summary:-

The authors have focussed the manuscript more on one question, it is now easier to tread and understand, although still not straightforward to read. I still had to read the paper numerous times to take it in.

For me it raises a couple of new questions because it is now a bit easier to follow.

They agreed that their interpretation of their findings was over reaching in terms of what their results show, however I still feel this manuscript overstates what conclusions can be drawn from the manuscript, and very much recommend modifying some of the sweeping statements made in the conclusion of the manuscript and abstract (see below)

They have included reporter assay work for one of the lung specific associations they found which validates their findings from publicly available transcriptomic data, this is a good addition to the paper.

I would still like the authors to highlight that the findings need more follow up and integration with other approaches to prove the link between gene expression in lung and other tissues, and conversion of t1d genetic risk.

Abstract:-

I dont agree with this sentence:- "Our analysis identified an eQTL between rs6679677 and AP4B1-AS1, in lung tissue, as the greatest genetic contributor to the risk of development of T1D."

I think the greatest genetic contributor to T1D risk is HLA genetic variation (I actually dont really know how they analysed the HLA region in this paper, see comment below). By what measure is AP4B1 the greatest genetic contributor to development of T1D? - I just dont see how their results support this statement. Do they mean that the greatest tissue specific impact of t1d associated SNPs (from the tissues that they studied at the development times they studied them) is the action of this SNP in lung tissue? Please can they clarify and put the result into an easier to understand context of t1d risk overall.

I don't understand this final sentence of the abstract : "Our results identify key genetic factors influencing the conversion of genetic factors into T1D risk and may help explain the reported association between respiratory infections and risk of islet autoantibody seroconversion reported in young children."

Suggest something like: Our results identify tissue specific eQTIs of SNPs associated with T1D, with the strongest tissue specific effects being in the lung. This tissue specific effect may help explain the association between respiratory infections and risk of islet autoantibody seroconversion in young children and warrants further mechanistic follow up.

The thread of the above statements being too general, over reaching, and not being focussed on their results and the limitations of what they have studied is also continued in their conclusion.

Methods

These are now clearer, thank you.

I have a question relating to HLA:-

They reference Gamazon, E. R. et al. (Using an atlas of gene regulation across 44 human tissues to inform complex disease- and trait-associated variation. Nat. Genet. 50, 956–967 (2018)) supplementary figure 14, in their discussion of their lung specific findings.

This Gamazon paper highlights a very large difference when HLA loci are included in the spatial analyses, and highlights the importance of HLA to T1D associated genetic risk. However the authors manuscript paper does not say if HLA SNPs are or are not included in their analysis, whether their inclusion or removal has an impact on their results, and seem to skirt over the fact that the vast majority of measurable genetic risk for t1d is associated with the HLA region, in contrast to their statement in the abstract and conclusions of the paper. This need to be rationalised. Is HLA risk is not mediated via a tissue specific eQTL, so that the majority of t1d genetic risk is not being studied, and the authors are studying only the t1d genetic risk that is tissue specific, this makes more sense to me, but if this is the case it needs to be clarified and used to moderate the conclusions.

Reviewer #2 (Remarks to the Author):

We appreciate that the authors have carefully considered our comments from the initial round of review. The quality of writing is now much more clear, and I currently understand the major points of this manuscript to be the following:

- (a) The group used a machine learning / data analysis framework leveraging eQTL and Hi-C resources to partition the tissue-relevance of T1D risk variants. Based on this framework, they found lung was the most disease relevant tissue in terms of the overlap between T1D SNPs and presence of eQTL effects.
- (b) They built on findings from the machine learning framework to constructed a risk score and demonstrated predictive value of the risk score in the WTCCC with an AUC of 0.76.
- (c) They validated the functional relevance of the allele specific enhancer activity for rs6679677 in lung epithelial cells.

With the better understanding of the overall message of this manuscript, I was now able to give the manuscript a more comprehensive assessment. Currently, I have major concerns about the manuscript which limit my overall enthusiasm about potential impact of the work. First, while the machine learning framework leveraging Hi-C and eQTL information is compelling, it is not clear how this framework contributes to polygenic / genetic risk prediction beyond current methods. I believe the partitioning the strength of spatial eQTL evidence for T1D risk SNPs across GTEx tissues is still potentially useful; I wonder if the authors may want to focus more on this aspect of the work.

Besides these overall points, I have detailed my other major points below.

Major:

1. In the Introduction of your manuscript, you suggest your proposed machine learning approach may provide a way to improve on prediction performance of polygenic risk scores (PRS). As you have noted, prior GWAS have already identified a large number of genetic variants related to T1D risk and progression. Thus:
 - a. When you present the predictive performance of your proposed methods (in Figure 3), you should also provide in the figure and/or text a comparison to the AUC obtained by using a polygenic risk score or genetic risk score constructed using standard methods and variants selected from GWAS of T1D. Prior studies of T1D PRS appear to achieve AUC > 0.8 for prediction of T1D risk, while your risk model appears to achieve AUC ~ 0.76. Thus, it is not clear how you would convince people to use your risk model rather than the existing PRS approaches.
 - b. It seems the risk prediction is emphasized a lot in the Introduction and Results. Why don't you say

anything about risk prediction in the Abstract?

2. Overall, the manuscript reasonably gives special treatment to the rs3087243-CTLA4 eQTL which “contributed an average of 4% to T1D risk.” Please explain in more detail whether and how you handled the HLA-region associations for T1D, as these also account for a very large proportion of T1D risk. Also, did you note any particular eQTL and/or Hi-C relationships for the important HLA-region variants?

3. p. 4, line 94: Please clarify how your Hi-C chromatin contact libraries were selected and curated for inclusion in this manuscript. Did you select Hi-C sources based on tissue of origin?

4. p. 4, line 96-97: Currently, you note a spatial eQTL is “an eQTL in which the expression quantitative trait is associated with a SNP and gene that are captured as physically interacting within the nucleus.” Based on your definition, I believe you mean that the eQTL variant should lie outside the genomic position boundaries of the gene. Could you please clarify in the text?

5. p. 5, line 104: Please clarify your technical working definitions of cis and trans associations. I assume it is based on cutoffs of genomic base pair distance between the SNP and gene positions, but it should be stated explicitly.

6. p. 7, line 144: You mention simulations here, but I did not see simulations described anywhere in the methods. What did you simulate?

Re: Decision on manuscript COMMSBIO-20-2098A-Z

Dear Referees

Thank you for your comments on our manuscript. We contend that our work is novel and our approach provides a significant advance over other previously reported methods. The novelty revolves around the ability of our method to: 1) rank the contributions that SNPs make to a phenotype through regulatory changes; 2) identify the tissues in which these changes are occurring; and 3) include effects from variants that do not have detectable eQTLs in the reference library that is used in the assay. The application of the method to T1D resulted in the identification of significant T1D associated eQTLs within the lung and may help explain the reported association between respiratory infections and risk of islet autoantibody seroconversion reported in young children. We have revised the text of the manuscript and clarified these points, in line with your comments.

Sincerely

Justin M. O'Sullivan

Reviewer 1

- 1) I would still like the authors to highlight that the findings need more follow up and integration with other approaches to prove the link between gene expression in lung and other tissues, and conversion of t1d genetic risk.
 - We have included the following statement at the end of the abstract:
“This tissue-specific effect may help explain reported associations between respiratory infections and risk of islet autoantibody seroconversion in young children and warrants further mechanistic follow up.”
 - in the discussion:
“The novelty of the approach we undertook lies in: 1) the integration of T1D-associated SNPs with their tissue-specific eQTLs (in both cis and trans); 2) interpreting individual case and control genotypes in terms of these tissue-specific eQTL effects; 3) including effects from variants that do not have detectable eQTLs in the reference library that is used in the assay; and 4) the application of machine learning to select and rank the tissue-specific eQTL effects that confer disease risk. This approach moves T1D research away from a candidate gene approach to include gene regulatory changes, including within the HLA locus, as possible contributors to the risk of developing T1D. However, all results are putative until they are followed up by integrative empirical methods that prove the link between gene expression in the lung and other tissues, and the conversion of T1D risk.”
- 2) Abstract I dont agree with this sentence:- “Our analysis identified an eQTL between rs6679677 and AP4B1-AS1, in lung tissue, as the greatest genetic contributor to the risk of development of T1D.”

I think the greatest genetic contributor to T1D risk is HLA genetic variation (I actually dont really know how they analysed the HLA region in this paper, see comment below). By what measure is AP4B1 the greatest genetic contributor to

development of T1D? - I just don't see how their results support this statement. Do they mean that the greatest tissue specific impact of T1D associated SNPs (from the tissues that they studied at the development times they studied them) is the action of this SNP in lung tissue? Please can they clarify and put the result into an easier to understand context of T1D risk overall.

- We apologise for this ambiguity. We have corrected the text throughout the manuscript to remove any reference to the 'greatest contributor to T1D risk'. With respect to the specific sentence in the abstract this now reads:
 - "In so doing, our analysis ranked the tissue-specific transcription regulatory effects for T1D-associated genetic variants and estimated their relative contributions to conversion to T1D. An eQTL (rs6679677) associated with changes to *AP4B1-AS1* transcript levels, in lung tissue, was identified as making the largest gene regulatory contribution to the risk of development of T1D."

- 3) I don't understand this final sentence of the abstract: "Our results identify key genetic factors influencing the conversion of genetic factors into T1D risk and may help explain the reported association between respiratory infections and risk of islet autoantibody seroconversion reported in young children."

Suggest something like: Our results identify tissue specific eQTLs of SNPs associated with T1D, with the strongest tissue specific effects being in the lung. This tissue specific effect may help explain the association between respiratory infections and risk of islet autoantibody seroconversion in young children and warrants further mechanistic follow up.

- We thank the reviewer for their suggestion. We have modified the end of the abstract to read:
 - "Our results identify tissue specific eQTLs for SNPs associated with T1D, with the strongest tissue specific effects being in the lung. This tissue specific effect may help explain reported associations between respiratory infections and risk of islet autoantibody seroconversion in young children and warrants further mechanistic follow up."
- 4) The thread of the above statements being too general, over reaching, and not being focussed on their results and the limitations of what they have studied is also continued in their conclusion.
 - We have modified the conclusion to read:
 - "In conclusion, our work provides novel insights into the role of variation in gene regulation in the risk of developing T1D. The transcriptional changes (including to *AP4B1-AS1* and *CTLA4*) we identified in the lung may help explain the reported association between respiratory infections and risk of islet autoantibody seroconversion reported in young children."

- 5) I have a question relating to HLA:-

They reference Gamazon, E. R. et al. (Using an atlas of gene regulation across 44 human tissues to inform complex disease- and trait-associated variation. *Nat. Genet.* 50, 956–967 (2018)) supplementary figure 14, in their discussion of their lung specific

findings.

This Gamazon paper highlights a very large difference when HLA loci are included in the spatial analyses, and highlights the importance of HLA to T1D associated genetic risk. However the authors manuscript paper does not say if HLA SNPs are or are not included in their analysis, whether their inclusion or removal has an impact on their results, and seem to skirt over the fact that the vast majority of measurable genetic risk for t1d is associated with the HLA region, in contrast to their statement in the abstract and conclusions of the paper. This need to be rationalised. Is HLA risk is not mediated via a tissue specific eQTL, so that the majority of t1d genetic risk is not being studied, and the authors are studying only the t1d genetic risk that is tissue specific, this makes more sense to me, but if this is the case it needs to be clarified and used to moderate the conclusions.

- We thank the referee for pointing out the omission of the HLA locus. We had included HLA variants in our models. We have modified the text to include a new section in the results on the regulatory changes we observed at the HLA locus.
 - “Regulatory changes in the HLA locus contribute to the risk of developing T1D in both models
The HLA-region is strongly associated with the development of T1D, accounting for 40-50% of the familial aggregation¹³. We observed spatial eQTLs involving HLA (Supplementary Table 13) within models 1 and 2. The significant HLA spatial eQTLs involved SNPs rs2251396-PSORS1C1, rs2251396-MICA, rs3129889-HLA-DRB5, and rs9268645-HLA-DQB2 and were observed as contributing to risk of developing T1D in multiple tissues (e.g. adrenal gland, transverse colon, small intestine, spleen, heart left ventricle, sun exposed skin, testis, and thyroid).”
- We have also modified the discussion to read:
 - “It is widely recognized that the vast majority of measurable genetic risk for T1D is associated with the HLA region and polymorphisms of the class II HLA DQ, DR, and DP genes¹³. Typically these effects are ascribed to polymorphisms within the HLA genes that affect the shape of the peptide binding groove and the scope of the peptides that can bind to the allele and thus be presented on the cell surface¹³. Our analysis focused on the identification and ranking of the tissue-specific regulatory changes and thus does not capture variants that change the function or structural properties of a gene encoded protein. Therefore, it is notable that the HLA locus variants that were retained within our models (rs3129889-HLA-DRB5, and rs9268645-HLA-DQB2) regulated expression of DRB and DQB alleles. In addition, rs2251396 was associated with expression of PSORS1C1, which has been associated with T1D²⁰. Similarly, rs2251396 was associated with expression of MICA, which has previously been identified as part of an extended HLA haplotype that associates with T1D risk²¹. Collectively these observations support the hypothesis that changes in HLA gene expression contribute to T1D risk, in addition to the recognized role for HLA polymorphisms.”

Reviewer #2 (Remarks to the Author):

- 1) Currently, I have major concerns about the manuscript which limit my overall enthusiasm about potential impact of the work. First, while the machine learning framework leveraging Hi-C and eQTL information is compelling, it is not clear how this framework contributes to polygenic / genetic risk prediction beyond current methods. I believe the partitioning the strength of spatial eQTL evidence for T1D risk SNPs across GTEx tissues is still potentially useful; I wonder if the authors may want to focus more on this aspect of the work.
 - We thank the referee for their suggestion and apologize for the confusion. As suggested, we have changed the manuscript to focus on partitioning of the spatial eQTL evidence for T1D risk SNPs.

- 2) In the Introduction of your manuscript, you suggest your proposed machine learning approach may provide a way to improve on prediction performance of polygenic risk scores (PRS). As you have noted, prior GWAS have already identified a large number of genetic variants related to T1D risk and progression. Thus:
 - a. When you present the predictive performance of your proposed methods (in Figure 3), you should also provide in the figure and/or text a comparison to the AUC obtained by using a polygenic risk score or genetic risk score constructed using standard methods and variants selected from GWAS of T1D. Prior studies of T1D PRS appear to achieve $AUC > 0.8$ for prediction of T1D risk, while your risk model appears to achieve $AUC \sim 0.76$. Thus, it is not clear how you would convince people to use your risk model rather than the existing PRS approaches.
 - As suggested by the referee, we have changed the focus of the manuscript to the partitioning of the spatial eQTL evidence for T1D SNPs. We have clarified our use of the model AUCs within the text as follows:
 - “In this study the AUC was used to identify the top performing predictor developed using the additive tissue-specific contributions of the spatial eQTLs within genotypes from individuals who developed T1D. As such, our predictor AUC is not directly comparable to AUCs generated by polygenic risk scores on the same datasets.”
 - “The mean AUC prediction, calculated from the 50 T1D regularised logistic regression predictors, was only 1.7% different (i.e. $[0.76-0.747]/0.76$) and the distribution encompassed the original AUC (min = 0.712, max = 0.771, standard deviation of 0.14; Supplementary Fig. 3). Therefore, we concluded that the predictor created with the optimized hyperparameters performed well across different data sets.”

- 3) It seems the risk prediction is emphasized a lot in the Introduction and Results. Why don't you say anything about risk prediction in the Abstract?
 - We have modified the manuscript to de-emphasize and remove mention of “risk prediction”.

- 4) Overall, the manuscript reasonably gives special treatment to the rs3087243-CTLA4 eQTL which “contributed an average of 4% to T1D risk.” Please explain in more detail whether and how you handled the HLA-region associations for T1D, as these also account for a very large proportion of T1D risk. Also, did you note any particular eQTL and/or Hi-C relationships for the important HLA-region variants?
 - We thank the referee for pointing out the omission of the HLA locus. We had included HLA variants in our models. We have modified the text to include a new section in the results on the regulatory changes we observed at the HLA locus.

- a. “Regulatory changes in the HLA locus contribute to the risk of developing T1D in both models
The HLA-region is strongly associated with the development of T1D, accounting for 40-50% of the familial aggregation¹³. We observed spatial eQTLs involving HLA (Supplementary Table 13) within models 1 and 2. The significant HLA spatial eQTLs involved SNPs rs2251396-PSORS1C1, rs2251396-MICA, rs3129889-HLA-DRB5, and rs9268645-HLA-DQB2 and were observed as contributing to risk of developing T1D in multiple tissues (e.g. adrenal gland, transverse colon, small intestine, spleen, heart left ventricle, sun exposed skin, testis, and thyroid).”
- We have also modified the discussion to read:
- b. “It is widely recognized that the vast majority of measurable genetic risk for T1D is associated with the HLA region and polymorphisms of the class II HLA DQ, DR, and DP genes¹³. Typically, these effects are ascribed to polymorphisms within the HLA genes that affect the shape of the peptide binding groove and the scope of the peptides that can bind to the allele and thus be presented on the cell surface¹³. Our analysis focused on the identification and ranking of the tissue-specific regulatory changes and thus does not capture variants that change the function or structural properties of a gene encoded protein. Therefore, it is notable that the HLA locus variants that were retained within our models (rs3129889-HLA-DRB5, and rs9268645-HLA-DQB2) regulated expression of DRB and DQB alleles. In addition rs2251396 was associated with expression of PSORS1C1, which has been associated with T1D²⁰. Similarly, rs2251396 was associated with expression of MICA, which has previously been identified as part of an extended HLA haplotype that associates with T1D risk²¹, Collectively these observations support the hypothesis that changes in HLA gene expression contribute to T1D risk, in addition to the recognized role for HLA polymorphisms.”
- 5) p. 4, line 94: Please clarify how your Hi-C chromatin contact libraries were selected and curated for inclusion in this manuscript. Did you select Hi-C sources based on tissue of origin?
- We have modified the text to read
- “The Hi-C libraries that were used in this study included immortalized cell lines and primary human tissues (Supplementary Table S2) and were chosen to ensure a range of known, possible interactions were included in the analysis.”
 -
- 6) p. 4, line 96-97: Currently, you note a spatial eQTL is “an eQTL in which the expression quantitative trait is associated with a SNP and gene that are captured as physically interacting within the nucleus.” Based on your definition, I believe you mean that the eQTL variant should lie outside the genomic position boundaries of the gene. Could you please clarify in the text?
- We have modified the text to read:
- “According to our definition, the eQTL variant can sit anywhere within the genome. This includes within the boundaries of the gene, as long as the gene is covered by ≥ 3 restriction fragments in the Hi-C library. This minimum connection distance is determined by Hi-C resolution, which cannot distinguish spatial connections between ligated contiguous restriction sites versus an undigested restriction site.”

- 7) p. 5, line 104: Please clarify your technical working definitions of cis and trans associations. I assume it is based on cutoffs of genomic base pair distance between the SNP and gene positions, but it should be stated explicitly.
- We have modified the text to read
 - “The eQTL-eGene interactions were categorized as either: cis, the eQTL and eGene are separated by a linear distance of $\leq 1\text{Mb}$ on the same chromosome; or trans, eQTLs and their eGenes were separated by $>1\text{Mb}$ on the same chromosome or located on different chromosomes. Notably, of the 256 T1D-associated SNPs with spatial-eQTLs...”
- 8) p. 7, line 144: You mention simulations here, but I did not see simulations described anywhere in the methods. What did you simulate?
- We have modified the text to read:
 - ” Tissue-specific contributions to the T1D risk were extracted from each of the 50 T1D regularised logistic regression predictors as the sum of the absolute values of the model weights associated with each tissue. We then ranked the tissue-specific contributions to the 50 regularised logistic regression predictors. This ranking identified the lung as the top average contributor to the relative risk (case:control) of developing T1D. Across all 50 regularised logistic regression predictors, the lung explained a mean of 13.6% (standard deviation of 2.51%) of the relative risk of developing T1D (Fig. 2; Supplementary Table S8).”

Reviewers' comments:

Reviewer #2 (Remarks to the Author):

In the current revision of the manuscript, the authors have focused more on the value of the Hi-C framework as well as the insights into tissue-specific that are gained from the framework. While I still find the manuscript somewhat difficult to read, I am now able to follow the main points. Further, the current version of the manuscript does make clear the value of the overall framework in combining Hi-C and eQTL information in identifying tissue-specific contributions to disease, particularly in pulling out trans-eQTL effects which are often difficult to incorporate in standard frameworks.

I have a few remaining comments / suggestions for the authors:

Major:

1. It appears a major value of using Hi-C information and spatial eQTL is in pulling out trans-eQTL which can often be difficult to interpret in the absence of chromatin information. Toward this end, it will be useful to state more explicitly when you are talking about cis versus trans eQTL effects. For example, please note in the main text whether the SNP rs6679677 is regulating AP4B1-AS1 and CTLA4 in cis or in trans, as this information will be of interest to readers. Similarly, other eQTL-eGene pairs should be labeled as to whether they are cis or trans relationships.
2. Your manuscript places a lot of emphasis on tissue-specific eQTL, which makes sense as GTEx provides a nice resource for this information. In contrast, it is not clear to me which tissues were used to extract chromatin interactions from Hi-C and whether you attempted to examine tissue specific effects to assign the eQTL-eGene interactions, or whether the Hi-C libraries were examined in aggregate only. I understand there is not much tissue specific Hi-C information available, but it would still be good to clarify this information.
3. Line 181, "We trained and validated the predictive accuracy of a multinomial logistic regression predictor..." – it is unclear why you would need multinomial logistic regression to predict disease case/control status since there are only two possible outcomes. Do you mean multivariate logistic regression?
4. Line 212: You compared model performance in subsets of WTCCC data to a model generated with the overall WTCCC and "concluded that the predictor created with the optimized hyperparameters performed well across different data sets". This exercise is not convincing or useful. If you want to examine model performance across different data sets, you should at least use cross validation rather than subsetting of your original data.

Minor:

1. Line 95, "this still did provide any significant information..." – I think you mean "did not"
2. Line 141, "We define a spatial eQTL as SNPs..." – I think you mean "a SNP" not "SNPs"

Re: Decision on manuscript COMMSBIO-20-2098A-Z

Dear Referees

Thank you for your comments on our manuscript. We have revised the text of the manuscript and clarified these points, in line with your comments. We have also modified the abstract and made minor changes to labelling (fig. -> figure etc...) to meet journal requirements.

Sincerely

Justin M. O'Sullivan

Reviewer2

Major:

- 1) It appears a major value of using Hi-C information and spatial eQTL is in pulling out trans-eQTL which can often be difficult to interpret in the absence of chromatin information. Toward this end, it will be useful to state more explicitly when you are talking about cis versus trans eQTL effects. For example, please note in the main text whether the SNP rs6679677 is regulating AP4B1-AS1 and CTLA4 in cis or in trans, as this information will be of interest to readers. Similarly, other eQTL-eGene pairs should be labeled as to whether they are cis or trans relationships
 - We have included a column in Supplementary Table 11 “Ranking of eQTLs on tissue-specific contribution to T1D risk using the final T1D classification model” to identify if the eQTL effects in the final regularized regression model (model 2) are acting in cis or trans.
 - We have modified the text at multiple places to ensure that the reader is aware the key interactions we discuss are occurring in cis. The text now reads:
 - *“The major contributor eQTL (rs6679677) down regulated AP4B1-AS1 transcript levels in cis (i.e. rs6679677 and AP4B1-AS1 are <1Mb apart) in the lungs”*
 - *“cis-acting eQTL involving rs6679677 as the largest single contributor of the risk for developing T1D”*
 - *“the rs3087243-CTLA4 cis-acting (i.e. <1Mb apart) spatial eQTL”*
 - *“The significant HLA spatial cis-acting eQTLs involved SNPs rs2251396-PSORS1C1, rs2251396-MICA, rs3129889-HLA-DRB5, and rs9268645-HLA-DQB2 ... “*
- 2) Your manuscript places a lot of emphasis on tissue-specific eQTL, which makes sense as GTEx provides a nice resource for this information. In contrast, it is not clear to me which tissues were used to extract chromatin interactions from Hi-C and whether you attempted to examine tissue specific effects to assign the eQTL-eGene interactions, or whether the Hi-C libraries were examined in aggregate only. I understand there is not much tissue specific Hi-C information available, but it would still be good to clarify this information.

- We have added the following statement in the paragraph “Identification of SNP-gene pairs and expression QTL associations in human tissues” within the Methods section to clarify this point.
 - *“These published libraries were chosen to ensure a wide-range of known interactions and were analysed in aggregate within the pipeline to address a lack of tissue-specific Hi-C, interactions captured within ‘relevant’ published Hi-C libraries.”*

- 3) Line 181, “We trained and validated the predictive accuracy of a multinomial logistic regression predictor...” – it is unclear why you would need multinomial logistic regression to predict disease case/control status since there are only two possible outcomes. Do you mean multivariate logistic regression?
- We thank the reviewer for pointing out this error. We have corrected the text to read “regularised logistic regression” throughout the manuscript.

- 4) Line 212: You compared model performance in subsets of WTCCC data to a model generated with the overall WTCCC and “concluded that the predictor created with the optimized hyperparameters performed well across different data sets”. This exercise is not convincing or useful. If you want to examine model performance across different data sets, you should at least use cross validation rather than subsetting of your original data.
- We thank the reviewer for pointing out the inaccuracy in our statement concerning the cross-validation of model 1.
 - The models in our manuscript were validated using both internal (model 1) and external cross-validation (model 2). The use of internal cross-validation is standard practice in machine learning (Ref: Raschka, S. Model evaluation, model selection, and algorithm selection in machine learning. *arXiv Prepr. arXiv1811.12808* (2018)) and showed that the predictor, which was fitted with optimised hyperparameters, generalises well across different subsets of WTCCC derived eQTL data. By contrast, the cross-validation of Model 2 on the UKBiobank data set demonstrated that the proposed model generalises well across different biological data sets.
 - We have addressed this in the manuscript by modifying the results paragraph to read:
 - *“Essential feature selection was performed using the Mann-Whitney U test⁸ (selected 2048 data features from 6307 eQTL features and 29 SNP features with unknown eQTL effects) and lasso regularization of the logistic regression. The regularization parameter was optimized by sampling 80% of the WTCCC derived eQTL dataset, identifying optimal hyperparameters from this sample, and evaluating the performance of the algorithm on the remaining 20% using the optimal hyperparameters. In order to identify optimal hyperparameters, the selected 80% of data were divided into 10 subsets with each subset having approximately the same number of samples. For every hyperparameter value, the prediction model was trained repeatedly on 9 subsets and evaluated on the remaining subset until every subset had been used for evaluation once. The optimal regularization parameter was selected based on the AUC (model 1). Applying this model to the validation data (20% of the WTCCC derived eQTL dataset) resulted in an AUC of 0.76, which is acceptable for our purpose of developing an interpretable machine learning model, which identifies the top performing predictors from additive tissue-specific contributions of the spatial eQTLs within genotypes from individuals who developed T1D. In order to quantify the uncertainty of the model performance subjective to different splits of the WTCCC data into training and test sets, we sampled 50 different training and test sets by repeating a 5-fold internal cross-validation of the WTCCC data a total of 10 times. This experiment generated 50 out-of-sample AUC values from 50 different T1D regularised logistic regression predictors created with model 1’s optimised hyperparameters. Each predictor was trained on a different subset comprising 80% of the WTCCC data and evaluated on the remaining 20% (Fig. 2; Supplementary Table 7; Methods). The 50 out-of-sample AUCs were varying between 0.712 and 0.771 with a mean of 0.747 and a standard deviation of 0.14 (Supplementary Figure 3). Internal cross validation is a standard practice in machine learning⁹ and showed that model 1 predictor, which was fitted with optimised*

hyperparameters, generalises well across different subsets of WTCCC derived eQTL data.”

- In addition, we have added the following sentence in the discussion.

The regularized logistic regression models used in this manuscript were validated by both internal cross-validation (model 1) which is standard practice in machine learning⁹ and external cross-validation using the different cohorts (model 2).

[Ref 9: Raschka, S. Model evaluation, model selection, and algorithm selection in machine learning. *arXiv Prepr. arXiv1811.12808* (2018)]

Minor:

- 1) Line 95, “this still did provide any significant information...” – I think you mean “did not”
 - We have corrected this error and replaced “did” with “did not”.
- 2) Line 141, “We define a spatial eQTL as SNPs...” – I think you mean “a SNP” not “SNPs”
 - We have changed the text as suggested.